# Neglecting Maternal Depression Compromises Child Health and Development Outcomes, and Violates Children’s Rights in South Africa

**DOI:** 10.3390/children8070609

**Published:** 2021-07-19

**Authors:** Kebogile Elizabeth Mokwena

**Affiliations:** Department of Public Health, Sefako Makgatho Health Sciences University, Molotlegi Drive, Ga-Rankuwa, Pretoria 0204, South Africa; kebogile.mokwena@smu.ac.za

**Keywords:** screening, maternal depression, child health, developmental outcomes, cognitive development, child mortality

## Abstract

The intention of the South African Children’s Act 38 of 2005 is to provide guarantees for the protection and promotion of optimum health and social outcomes for all children. These guarantees are the provision of basic nutrition, basic health care and social services, optimal family or parental care, as well as protection from maltreatment, neglect and abuse services. However, despite these guarantees, child and maternal mortality remain high in South Africa. The literature identifies maternal depression as a common factor that contributes to negative health and social outcomes for both mothers and their children. Despite the availability of easy-to-use tools, routine screening for maternal depression is not carried out in public health services, which is the source of services for the majority of women in South Africa. The results are that the mothers miss out on being diagnosed and treated for maternal depression, which results in negative child outcomes, such as malnutrition, as well as impacts on mental, social and physical health, and even death. The long-term impacts of untreated maternal depression include compromised child cognitive development, language acquisition and deviant behaviors and economic disadvantage in later life. The author concludes that the neglect of screening for, and treatment of maternal depression therefore violates the constitutional rights of the affected children, and goes against the spirit of the Constitution. The author recommends that maternal and child health services integrate routine screening for maternal depression, which will not only satisfy the Constitutional mandate, but also improve the health and developmental outcomes of the children and reduce child mortality.

## 1. Introduction

The South African Constitution [1], the supreme law of the country, is a gold standard that is appropriate for use to assess the extent to which the government upholds the rights of its citizens through providing basic services to all citizens, and services to restore or uphold their human dignity. While social and economic inequality is a feature of the global society, the purpose and spirit of the Constitution is to provide services that bridge the gap between the deprived and those that have resources and power. In other words, the Constitution guarantees the rendering of basic services for all citizens, irrespective of their economic strengths or social standing. The Constitution therefore addresses social rights even in the field of poverty and welfare for all citizens [1,2], and can only provide these guarantees in the context of the prevailing circumstances at a given time.

The purpose of the Children’s Act 38 of 2005 intends to give effect to a number of rights of children, as contained in the South African Constitution. The purpose of this paper is to use the literature to highlight the impact of maternal depression on the health and social wellbeing of the child, and to argue that neglecting to address maternal depression violates the rights of the children of these mothers. Specifically, the Children’s Act 38 of 2005 guarantees to support the early childhood development of children by providing basic nutrition, basic health care and social services, optimal family or parental care, as well as protection from maltreatment, neglect and abuse [3]. Maternal depression is a mental condition that, if left undiagnosed and untreated, works against every one of those aspects of child health and development. This paper uses a literature review to argue that neglecting to diagnose and treat maternal depression compromises current and future health, developmental and social outcomes of the affected children, and therefore violates their constitutional rights.

### 1.1. Methodology

Literature searches from Pubmed, ScienceDirect and Google Scholar were used to identify studies relating to the impact of maternal mental health on child health outcomes. Key words used included antenatal depression, postnatal depression, child health outcomes, child developmental outcomes and child mortality in developing countries, Africa and South Africa. Within the South African Constitution, the South African Children’s Act 38 of 2005 was reviewed for its intended guarantees on children’s rights. The negative impacts of maternal mental health on child outcomes were aligned against the aspects of the Children’s Act 38 of 2005, which were violated by the failure of the health system to screen for maternal depression. The recommendations were drawn from the discussion of the overall review of the literature.

### 1.2. Prenatal and Postnatal Depression

When maternal depression occurs prenatally, the children are disadvantaged even before they are born, which therefore affects their early development. Maternal depression during pregnancy has been associated with low birth weight [4], intrauterine growth restriction [5,6,7], premature delivery [8,9] and poor breastfeeding outcomes [10,11], which in themselves are risk factors for a range of health and developmental challenges. Postnatal depression manifests after the child is born and it manifests mostly as the social barrier to how the mother bonds and nurtures her child. Studies report that prenatal depression extends to the postnatal stage, with a compounded impact.

Whether maternal depression occurs before birth, after birth or during both periods, it interferes with the interaction between the mother and her child [12]. Literature confirms the increase in both pre and postnatal depression, especially in developing countries, including South Africa. This paper therefore refers to maternal depression, irrespective of when it occurs. A number of studies conducted in South Africa reported an increasing prevalence of maternal depression over the years, and these include the 49.3% reported in 2014 [13], 50.3% in 2015 [14] and 57.14% reported recently [15]. Although these studies were conducted in pockets of communities, and do not reflect national prevalence, they do confirm the seriousness of maternal depression in South Africa. What is concerning is that the public health care system in South Africa does not routinely screen for maternal depression, and that implies that many women with maternal depression are not diagnosed and/or treated. While this is an indictment of the system on the mental health of women, it is even a greater indictment of the system that is expected, but fails to promote the health and social outcomes of their children.

### 1.3. Maternal Depression and Child Health and Developmental Outcomes

As the global prevalence of mental disorders continues to increase, maternal mental disorders, which include pre and postnatal depression, carry a significant proportion of that increase [12]. This particularly applies to low- and middle-income countries, where the prevalence is not only higher [16], but resources to address the scourge are severely constrained [17,18]. This implies that many women who have maternal depression are not screened, diagnosed or treated, with resultant negative health and social outcomes for both the mother and the child. The situation continues to frustrate efforts to improve maternal and child health outcomes. It is for this reason that an urgent need for improved mental health care has been identified [19].

South Africa did not reach the Millennium Developmental Goals 4 and 5, which are about reduction in both child and maternal mortality [20], and the country is likely to miss meeting related Sustainable Developmental Goals, unless specific plans are developed and implemented to mitigate not only maternal and child mortality, but also to improve the health and social outcomes of the children [21]. Such specific plans were missing in the MDG era, and a key to the required improvement is to focus on factors which impact on both maternal and child wellbeing, and maternal depression is in that area. After all, maternal depression perpetuates low socio-economic status [22], which is a risk factor for negative health outcomes of both the mother and the child.

The rates of maternal depression are reported to be higher in low- and middle-income countries (LAMICs), where nearly 90% of the world’s children live, which means that the rights of millions of children living in these countries are being violated by disregarding the screening, diagnosis and treatment of their mothers’ depression [23]. This compromises the development of children in these areas [24], where the social environment presents additional socio-economic challenges against the optimum development of such children. Maternal depression has been associated with a range of long-term poor physical health conditions in childhood, as well as compromised mental and social functioning in young adults [25]. There is, therefore, a need to pay attention to maternal health in LAMICs, as this can contribute to important public health gains for the mother and her children.

Although maternal depression is a huge public health problem in low- and middle-income countries (LAMICs), including South Africa, it is largely neglected [26], despite enormous evidence that if not diagnosed and treated, maternal depression has long-term impacts on the social, mental and physical wellbeing of their children. Because the burden of maternal depression in developing countries is often hidden, it does not get the attention it deserves [27], probably because countries have not developed a political will to confront the situation. Maternal depression therefore remains a major source of disability which compromises their nurturing processes, which is essential for positive parental outcomes.

Although the child development process occurs over a prolonged period, children’s vulnerability or potential are mostly impacted upon by the first 3 years of life. It is during this time that attention should be given to afford the child the greatest support to enable him/her the best possible outcomes in various aspects of development [28]. Neglecting to meet the needs of the child during this crucial time may have long-term negative consequences which may be difficult to reverse. Studies conducted in high-income countries report that the impact of maternal depression can actually be intergenerational [23], which elevates the relevance of its prevention and treatment to human rights level. Studies on the effects of prenatal maternal depression on a developing fetus and child indicate that maternal depression during this period may be more significant as it sets the child back even before birth. Public health assessment of child risks associated with maternal depression therefore becomes one of the most important intervention to influence child health outcomes.

In South Africa, mothers who are depressed are mostly profiled by poor socio-economic status, as depicted by a low level of education, lower incomes, unemployment, problematic drinking of alcohol, having experienced interpersonal partner violence and having experienced food insecurity [29]. Moreover, inadequate access to health services is a feature of low socio-economic status, which puts children living in such environments at risk of a range of adverse childhood experiences, with resultant health problems that can transcend adulthood for several generations [30]. Maternal depression is therefore a public health concern with significant negative implications for child functioning, including the development of negative child effects and risk for later depression.

### 1.4. Impact of Maternal Depression on Breastfeeding

Breastfeeding is one of the key contributors to child health because its benefit to the child is not only nutritional, but also promotes the interaction between the mother and her child. However, maternal depression is negatively associated with breastfeeding practices because most women with postnatal depression are not able to do so, because of difficulties related to milk production and their poor mental health [31].

Mothers experiencing depression will therefore need to be enabled to initiate breastfeeding early and also to continue longer with breast feeding [32]. This cannot be implemented if the depression has not been diagnosed and treated, and thus robs the child of an opportunity to promote greater developmental support for maternal nutrition and breastfeeding [33]. Moreover, depressed mothers who wish to breastfeed but are not able to often perceive themselves as ‘failures’, and this can trigger mental health problems [34].

### 1.5. Maternal Depression and Child Malnutrition

Despite high prevalence of both maternal depression and child malnutrition in developing countries, both have not received adequate attention. Health literature reports a strong association between maternal depression and child malnutrition, which compounds the impact of the combination of the factors, more than if each one is considered separately. On the other hand, malnutrition is a primary cause of child morbidity and mortality, which increases health risks for such children [35].

The consequences of childhood malnutrition remain a major disabling factor in child health and developmental outcomes, and have contributed to increased hospitalization burden [36], as well as treatment at out-patient clinics [37]. Hospitalization is often because of acute infections such as diarrhea and chest infections [38,39]. Moreover, maternal depression and child malnutrition have been associated with child behavioral problems, attention deficits and poor cognitive function [40]. As an off-shoot of maternal depression, child malnutrition remains a major challenge for improving child health outcomes; thus, emphasizing the need to integrate maternal mental health services into the management of acute malnutrition programs [41].

### 1.6. Physical Health and Development

Maternal depression has also been reported to impact on the physical health of children [42], with a resultant higher risk of hospitalization [36]. Studies have also explained the poor health of children born from mothers who are depressed to be the result of reduced adaptive immunity in infancy [43,44,45] and allergic reactions [46], though the process needs further investigations. Maternal depression is also a risk factor for mothers failing to complete the immunization schedules, thus causing the child to be vulnerable to infectious diseases such as acute or chronic diarrhea [47]. Because of the depressed mothers’ compromised skills and probable low socio-economic status, maternal depression has also been associated with child malnutrition [48,49,50], which in itself is a risk factor for other infections and poor physical development.

Maternal depression has also been associated with physical growth impairment [51], which includes being underweight in early childhood, stunting [52] and child obesity [53,54,55]. There is thus the need to identify, treat and prevent maternal depression, which may help to reduce all these nutritional childhood diseases, especially in developing countries where child growth is commonly compromised. The combination of maternal depression and severe nutrition problems results in poor infant growth, which is more prevalent in developing countries [56].

### 1.7. Maternal Depression Impacts on Child Mental Health

Maternal prenatal depression has been associated with offspring depression across childhood and early adulthood [57,58], and the malnutrition experienced in childhood is a risk factor for depression when the child is in their teens [59]. Children of depressed mothers are exposed to both maternal psychopathology and other risks that are associated with maternal mental disorders [60], and these have been reported to persist in late childhood [61], the adolescent stage [62] and even the stage when such a child transitions to adulthood [63].

Studies have reported that maternal depression impacts child psychopathology across the first decade of life, and thus significant harm is perpetrated during this time which is crucial for child development [64]. Acknowledging this significant impact by treating mothers with depression will result in beneficial effects for such children. In a country such as South Africa where mental health resources are limited, the offspring may not be diagnosed or treated either, and the cycle of deprivation continues.

### 1.8. Maternal Depression Disrupts the Social Development of the Child

Several studies have reported strong associations between maternal depression and a number of social, physical and mental adverse outcomes in their children. Such impairment can occur during pregnancy and/or after birth [65].

Impaired social measurements include challenges in behavior, socioemotional adjustment and emotion regulation, and these can occur because of placental function which affects the child and stress reactivity during pregnancy [25]. Disrupted social development affects the overall wellbeing of the child and how he/she deals with the social environment. Furthermore, compromised social development negatively impacts on behavioral development, especially during pre-school years, which impacts on their schooling and career development [66]. Maternal depression is also a risk factor for antisocial behavior later in the life of the offspring [67], as well as a range of risky behaviors in early adolescence [68], which are serious situations that need to be prevented at all costs.

Although supportive relationships with caregivers have been reported to mitigate negative social outcomes [69], the availability of such supportive relationships cannot always be provided, which leaves the children to be socially and mentally exposed to developmental risks and outcomes. 

### 1.9. Cognitive Development 

Maternal depression is an example of adversity experienced in childhood, which affects brain and cognitive development and confers risk for pathology [70,71,72]. Other studies have associated a range of maternal mental health difficulties with children’s cognitive or academic achievement [73,74], especially in mathematics [51] and reading scores [75]. On the other hand, poor cognitive skills are linked to mental health problems, which results in a vicious cycle.

Maternal depression impacts on the language development of the child [76], which affects social and educational progress, because language and communication is a key tool in cognitive development. Furthermore, not only does language acquisition influence a child’s readiness for school, it also determines the child’s performance at school entry [77].

Moreover, the depressive symptoms identified among children whose mothers are depressed further suppress the child’s cognitive development, and result in a compounded impact.

### 1.10. Compromised Emotional Attachment to the Child

Maternal depression has a negative impact on the socioemotional outcomes of the child [70,71], and this compromises the emotional development of the child. Depressive mothers interact less with their children, show less love and affection and often miss symptoms that are contrary to the child’s development and wellbeing, thus impacting on the future development of the children. Children with compromised emotional attachment show attachment insecurity, delayed language acquisition, cognitive impairment, delay in achieving developmental milestones and low self-esteem [78,79]. Moreover, maternal depression and resultant emotional detachment is a risk factor for child abandonment [80,81], which increases the risk for poor outcomes exponentially.

### 1.11. Child Maltreatment Often Emerges from Maternal Depression

Maternal depression has been associated with child maltreatment [82,83,84]. On the other hand, children with a history of maltreatment are at risk of poor academic performance, compromised mental wellbeing and unbecoming behavior [85]. Maternal depression is also a risk factor for exposure of the child to trauma, the impact of which is similar to child maltreatment [86], with potentially serious psychological and developmental consequences which present as trauma [87]. Moreover, untreated maternal depression is a risk factor for psychological and physical aggression perpetrated on their children [84]. This highlights the importance of early detection and treatment of depression as a tool to prevent child maltreatment [88].

### 1.12. Maternal Depression and Child Mortality

Due to the risk of compromised immunity of children born from depressed mothers, the mortality risks for such children are increased, which is worsened by the poor parental skills of the mothers. Other studies have reported a strong association between maternal depression and child mortality [89,90]. Moreover, because of the poor mental health status, maternal depression has been associated with a significant risk for suicide or infanticide [91], which are the most severe outcomes of maternal depression [92,93]. The implications are that several children’s deaths continue to occur because of failure to diagnose and treat their mothers for depression. Because child mortality is often credited as a surrogate marker for the quality of care within a health service [94], the unacceptably high child mortality in South Africa requires a relook at other aspects of health services, which includes screening for maternal depression.

### 1.13. Economic Impact on the Mother and the Child

Maternal depression is not only a source of disability among women, but it contributes to economic and human costs, both for the mother and for her child in the future [70], thus resulting in economic deprivation. Early childhood development predicts future social capital and national productivity, hence the need to invest in this period of childhood by addressing maternal depression, which has the capacity to destroy all that social capital for the future. Moreover, the various health challenges that are associated with maternal depression create an economic burden. This is why the neglect of diagnosing and treating maternal depression is a violation of human rights, because it deprives the mother available treatment, while on the other hand, compromising the future economic prospects of the child, thus causing the monetary cost of neglecting psychosocial risk factors for both mothers and children to be significant [28].

### 1.14. Availability of Screening Tools for Maternal Depression

A number of instruments are available for screening for maternal depression, and these include the Self Reporting Questionnaire (SRQ) [95,96], the Zung Self-Rating Depression Scale [97,98], the Postpartum Depression Screening Scale (PDSS) [99,100] and the Edinburgh Postnatal Depression Scale (EPDS), which is probably the most commonly used because of its substantial sensitivity and specificity as a screening tool for maternal depression. It also has a high positive predictive value, which is the probability that women who are identified with high depressive symptoms are truly depressive on clinical assessments. The tool is also easy to use, and can be administered within a few minutes [101]. Despite the availability of these relatively easy-to-use tools, routine screening for maternal depression remains a dream, despite the heavy price paid by the children whose mothers are not diagnosed and/or treated. Hence, screening for maternal depression fails to occur, not because of the lack of easy-to-use tools, but apparently because of a lack of political will to save the lives of women and their children.

## 2. Recommendations

Based on the impact of neglecting maternal mental health on children’s rights, and in the context of the South African Primary Health Care system, the author makes the following recommendations: That maternal and child health services integrate systematic screening for pre and postnatal maternal depression, which is feasible [26]. This is especially appropriate in South Africa as a low-resource setting [90], with many people having a low socio-economic status, which is a risk factor for maternal depression [91]. The routine screening will come with the following benefits:It will enable access to mental health services for a significant proportion of women [36].It will provide statistics which can assist in estimating the burden of mental disorders among women of child-bearing age.It is a viable strategy to prevent infant and child morbidity and mortality [90], especially in South Africa, where maternal depression, maternal deaths and child mortality are high.That the ward-based outreach teams that are already part of the Primary Health Care community health centers and clinics be strengthened by providing training for a specific cadre of community health workers to provide this service [101]. Referrals to higher levels of care are to be integrated into the screening program. The ward-based outreach teams should be trained to facilitate the formation of self-help groups in communities, to offer ongoing social support.That screening for maternal depression be included in the prescribed minimum benefits in both private and public health services, to ensure that all mothers have access to this service.That specific training on the maternal screening program be provided through an accredited training service provider, and be funded from the national health budget.

## 3. Conclusions

Failure to screen for and treat maternal depression is a neglect of the state’s responsibility as the corporate parent, and represents an issue of social injustice [2]. The good intention of improving maternal and child health outcomes will not be realized until very specific interventions are implemented, and screening for maternal depression is one significant intervention to reach that goal. Failing to screen for maternal depression results in the violation of the rights of the children, with long-term and mostly irreversible outcomes. The social support needed to improve maternal mental health status can be implemented through the already existing Primary Health Care ward-based outreach teams, which have been established in Primary Health Care facilities.

## Data Availability

Not applicable.

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
