# Peer review of "Neglecting Maternal Depression Compromises Child Health and Development Outcomes, and Violates Children’s Rights in South Africa"

_children, 2021, doi:10.3390/children8070609_

Round 1

Reviewer 1 Report

The topic of the paper is important, and much information is given regarding the negative effects of maternal depression for the children, and regarding the needs to detect maternal depressions and give adequate treatment and help, supported by an extensive list of references.

However, the paper jumps from the description of the constitution to the topics and recommendations without describing any national health policy documents or national guidelines for the health services regarding such health services, or lack of such documents. This would be important to understand where the paper and its recommendations will fit in.

This also raises the question who is giving these recommendations. Are the recommendations the personal opinions of the author as a single person, or are they given on behalf of an organization? Usually such recommendations are given from an government or other agency, organization or collaboration and developed by a group of experts or stakeholders. 

Another related question is what type of paper this is. On page 2 line 7 it is stated that the paper 'uses literature review'. However, there is no information on methods used for literature searches, screening and selection of papers, quality assessment of papers or other issues which are now standards for systematic reviews or other forms of literature reviews. Methods used should be described in the paper (and possibly details or searches or other parts in an online attachment), or the author could remove or moderate the statements that the paper uses literature review. 

The paper does not claim to be clinical or health policy guidelines. There are also standards for guidelines, and such guidelines may be assessed using the tool AGREE II, which also indicates some standards for guidelines. 

Another type of paper this could be, is a position paper. This is a type of paper with less defined standards. However, such papers are also usually written on behalf of an organization or group.  

The strength of the paper and most of the content is regarding public health with focus on maternal depression and its consequences for children. There is limited information on the health services and how they attempt to meet these needs, and services that have been successful and that may be good examples to learn from. Some of the recommendation mention health services (with references) which could have been part of more detailed descriptions of services earlier in the paper.

A section on the health services in the main part of the paper would have contributed to a better basis for specific recommendations at the end of the paper and improvement of the recommendations. Such a section should also have included presentations and discussion of well documented methods to identify and treat maternal depression. Some such references are already in the paper, but the text gives limited information. 

Another alternative could be to let this be a public health paper ending with a summary of unmet needs for health services without specifying how these services should be given. Specific recommendations may also have greater impact on the services if given by governmental and/or other national relevant organizations representing relevant stakeholders.

The main structure of the paper according to the headings and subheadings seems adequate, unless the paper should follow any standards like those mentioned above or other standards. However, a problem is that some issues are mentioned in several sentences in different parts of the paper (e g  mentioning both prevalence, needs, lack of services and consequences for one type of problems). The paper would be improved by identfying such repetitions and keeping each in the primary paragraph on this issue, and controlling a logical sequence of paragraph within each topic. 

Mostly statements and claims are supported by references. Howevere, there are also many setences with statements or claims without any references. Where such sentences are an introduction to a paragraph containing more details with references, this may be formulated.

Regarding language, the paper is well written, except for a few places where there seems to be a word missing (e g 'perpetuates low socio economic ????' on page 3 line 1) og a word or two  too much (e g 'reduction of child and reduction of maternal mortality' on page 2 line 6 from bottom, whic probably should be 'reduction of child and maternal mortality').

Reviewer 2 Report

In this paper, Kebogile Mokwena examined  the effect of neglecting maternal depression on child health and development outcomes. The present study enquires a topical subject in psychiatry, as maternal depression is a very common mental health problem, and it could have a strong impact on the health and social wellbeing of the child.

The manuscript is consistent within itself and has a pleasant logic flow. The Introduction section is clear, providing the reader with concise information needed to understand the article and contextualize the research question. References are appropriate. I highly appreciated the Recommendations section and the purpose to give the reader a global picture of the topic.

Nonetheless, there are some minor issues that I wish the authors would fix to further improve the value of the article.

  1. Although it is not a systematic review, I suggest including a brief paragraph about the search methods, reporting a list of the themes explored by the authors.
  2. Some of the paragraphs could be merged to give a better flow to the paper (e.g., Maternal depression impacts on child mental health and Compromised emotional attachment to the child).
  3. In my opinion when offering Recommendations, it is useful to include a brief explanation on the methods. The authors could better contextualize this paragraph, describing how he came to these recommendations and whether other systematic recommendations exist in the literature.

Round 2

Reviewer 1 Report

The author has addressed all my comments adequately by correcting some specific details, added a brief paragraph on methods, given a short introduction to the recommendations, and given adequate responses to some other comments which the author did not find appropriate to follow.

Author Response

The comments of the reviewer confirm that the author responded adequately to previous comments, and do not require any additional modifications or amendments